# Mechanical Properties and Fire Resistance of Magnesium-Cemented Poplar Particleboard

**DOI:** 10.3390/ma12193161

**Published:** 2019-09-27

**Authors:** Nihua Zheng, Danni Wu, Ping Sun, Hongguang Liu, Bin Lou, Li Li

**Affiliations:** Wooden Material Science and Application, Beijing Forestry University, Beijing 100083, China; zheng_nihua@163.com (N.Z.); 15393366580@163.com (D.W.); sp1996sghr@163.com (P.S.); luobincl@bjfu.edu.cn (B.L.); bjfulili@126.com (L.L.)

**Keywords:** magnesium composites, fire resistance, mechanical properties, rice straw, particleboard

## Abstract

Magnesium-cemented poplar particleboard (MCPB) is a new environmental inorganic magnesium cementitious composite without formaldehyde release. In this study, soybean flour (SM), silane coupling agent (KH560), soybean protein isolate (SPI), polyethylene glycol (PEG-400), maleic anhydride (MAH), and polyacrylic acid (PAA) were added to MCPB to improve the mechanical properties, waterproofing properties, and flame retardancy. The results show that the SPI group had the best mechanical performance; the modulus of elasticity (MOE) was up to 5192 MPa and exceeded the Chinese national standard GBT 4897-2015 (GBT) by 67.4%, the modulus of rupture (MOR) reached 17.72 MPa and exceeded GBT by 18%. Thickness swelling (TS) after 24-hour immersion was 0.29% and reached the standard of GBT (≤16%). The fire resistance test of MCPB indicated that the KH560 group improved the flame retardancy, the heat release rate (HRR) was 18kW/m^2^, the total smoke produced (TSP) was 0.192 m^2^, the total heat release (THR) was 29.71 MJ/kg, which reached the Chinese national standard GBT 8624-2012.

## 1. Introduction

Magnesium cement poplar particleboard (MCPB) is made of poplar shavings and magnesia cementitious materials. Magnesia cementitious material (MC) is a kind of inorganic adhesive which consists of light burned magnesia powder (MgO), H_2_O and magnesium chloride, having a good compressive strength with crystal structure of 5Mg(OH)_2_.MgCl_2_.8H_2_O (5-phase, 5.1.8 phase) and 3Mg(OH)_2_.MgCl_2_.8H_2_O (3-phase, 3.1.8 phase). Poplar is a tree belonging to a broad-leaved wood type, fast-growing and high yielding, which has been widely planted in China. In recent years, due to the shortage of wood, poplar has gradually become the raw material of wood-based panels and for use in the artificial furniture industry. Poplar particleboard has advantages of light weight, good elasticity, and easy processing. The common particleboard with phenolic and urea formaldehyde contains formaldehyde to some extent. As reports go, formaldehyde-based adhesives will be consumed at about 20 million tons around the globe in 2019, primarily in the wood-processing industry [1], but which are harmful to human health [2]. MCPB is an eco-friendly material without formaldehyde and with good performance of mechanical properties, water resistance, and fire resistance compared with the common poplar particleboard.

Scholars have studied the production and modification of poplar particleboard such as formaldehyde release, waterproofing, flame retardancy, and heat preservation etc. Robert and Bronislaw used zirconia, slag, and ceramics to add to the magnesia refractory production process. The results showed that the addition of ZrO_2_ can increase the fire protection function and improve the corrosion resistance of materials [3]. Morteza studied the possibility of producing particleboard from waste cotton stalks, the results showed a perfect agreement between the estimated values and observed data, with an increase in the ratio of melamine-formaldehyde resin to urea-formaldehyde, shelling ratio while the amount of poplar in the core layer of the panels, the modulus of rupture (MOE), the modulus of elasticity (MOR), and the internal bonding (IB) all increased [4]. Seyedeh evaluated the potential use of amino silane coupling agent (SiNH) to improve the physical and mechanical properties of Urea formaldehyde (UF) -bonded poplar wood particleboard. The results showed all board properties were improved by the addition of silane coupling agent [5]. Seyedeh and Kazem used silane coupling agents to improve the bonding ability between wheat straw particles and UF resin, and they investigated surface properties (wettability and surface roughness) and hardness of particleboard made from UF-bonded wheat straw combined with poplar wood as affected by silane coupling agent content and straw/poplar wood particle ratios. The results showed that panels with higher amounts of silane had lower root mean square roughness (Rq) values [6]. Wang studied how to transform construction wood waste by magnesia-phosphate cement (MPC) into cement-bonded particleboards. The research demonstrated that red mud and wood waste were sustainable materials for producing particleboards and improved short-term water resistance [7]. Julio used the packaging of cement as an option for the manufacture of particle board. The results revealed that the packages have a high cement content and low lignin pulp. Furthermore, the increase in density reduced the physical properties while enhancing the mechanical properties [8]. Wan investigated the mechanical strength and dimensional stability of steam pre-treated banana trunk (free from synthetic adhesive). The findings of this study showed that the banana trunk waste had potential to be used as raw material for the manufacture of binderless particleboard, which can be used in green buildings [9]. Monteiro showed that wood particleboards bonded with sour cassava starch can display low density and good mechanical performance, the research optimized the pressing conditions of the production of panels [10]. Paiman researched treatment with vegetable oils on imparting better dimensional stability to wood and wood-based composites. The results revealed that, in comparison with untreated particleboard, the treated particleboard exhibited better thickness swelling (TS) and water absorption (WA) [11]. Ulrich found impregnation in alkyl ketene dimer (AKD) to impart urea- formaldehyde (UF) bonded particleboards with higher and longer-lasting hydrophobicity [12]. Wang and Yu studied two fast-growing species fir and poplar, in order to find compatibility with Portland cement in processing cement particleboard. The research found that Chinese fir had little inhibition on cement hydration [13]. An and Ronald studied the impact characteristics of wood- cement particleboard (WCPB) made with wood particles produced from waste streams. The tests indicated that WCPB had the capability of absorbing large amounts of energy, could minimize impacted vehicle decelerations, and reduced the potential for passenger injuries [14]. Raul and Alain researched gypsum particleboard reinforced with Portland cement, to give a composite with higher mechanical properties and an acceptable resistance to moisture. It can be concluded that Portland cement is a suitable reinforcing material for improving the performance of gypsum particleboard [15]. Li studied the hot-pressing technology of cement bamboo particleboard, using variance analysis and regressive analysis to determine the optimizing parameters [16].

In this research, poplar particleboard and magnesia cementitious material were mixed to prepare magnesium cement poplar particleboard (MCPB). Mechanical properties, water resistance, and fire resistance were tested and compared by adding various chemical additives, such as soybean flour (SM), silane coupling agent (KH560), soybean protein isolate (SPI), polyethylene glycol (PEG-400), and polyacrylic acid (PAA), respectively. By compared the modulus of rupture (MOR), modulus of elasticity (MOE), and water absorption thickness swell (TS), the final optimized formula was obtained [17]. The micro-structure of the composite was observed by field emission electron scanning microscopy (FESEM). Fourier transform infrared spectroscopy (FTIR), X-ray diffraction (XRD), and X-ray photoelectron spectroscopy (XPS) were used to research the chemical reaction, functional groups, and crystal structure. Finally, the fire resistance and thermal stability of MCPB were investigated by thermogravimetric analysis (TG) and cone calorimetry (CONE).

## 2. Materials and Methods 

Poplar shavings, density of air-dried 0.386 g/cm^3^, moisture content 10%–12%, irregular strips of 50 mm × 2 mm × 1 mm, were purchased from Hebei WenAn Wood Factory. Light burnt magnesium (MgO content 86.26%, molecular weight 40.31), magnesium chloride (MgCl_2_ content 98.3%, molecular weight 95.21), H_2_O, NaOH, silane coupling agent (KH560), maleic anhydride (MAH), polyacrylic acid (PAA), polyethylene glycol (PEG-400), soybean meal (SM, with 25% SM powder and 75% H_2_O), soybean protein isolate (SPI, with 12% SPI powder and 88% H_2_O) as listed in Table 1.

In this study, the molar ratio of MgO/MgCl_2_ (M value) was 5, and the molar ratio of H_2_O/MgCl_2_ (H value) was 15. The proportion of poplar shaving was 30% of the weight of MCPB. The proportion of chemical additives was in weight ratio of magnesium cement material (MC).

### 2.1. The Preparation of MCPB

First, MgCl_2_ was slowly added into distilled water to prepare brine (as Figure 1 shows), and stirred for about 1 min. The MgO powder was added into the brine and stirred for 3 min as fast-slow-fast, that is 1 min slow (72 rpm), 1 min fast (120 rpm) and 1 min slow again (72 rpm). Then the chemical addition was made and stirred for 3 min as before. After mixing with wood shavings, the mixture was spread on a tray (50 mm × 50 mm) with a mould (37 mm × 37 mm × 25 mm). After prepressing, the mould was removed and two steel bars (40 mm × 12 mm × 12 mm) were placed on both sides of the mixture to control the thickness. Finally, the material with the tray was put in the cold holder (CGYJ-100, Frequency Machinery Co., Ltd., Shijiazhuang, China) with 3.5 MPa pressure for 8 h. The MCPB needed to be maintained at a constant temperature in a humidity box (DHS-225, Beijing North Lihui Testing Equipment Co., Ltd., Beijing, China) for 14 days. The temperature and humidity were 22 °C and 65%, respectively. The steps of preparation were according to the China national standard GB/T 175-2007 [18]. 

### 2.2. Mechanical Property Test

According to the Chinese national standard GB/T 17657-2013 [19] and GB/T 17671-1999 [20], MCPB was sawn into 250 mm × 50 mm × 12 mm pieces. The modulus of elasticity (MOE) and the modulus of rupture (MOR) were tested in the mechanical testing machine by the three-point bending method. The methods for MOR and MOE are shown in Equations (1) and (2).
R = (3F × L)/(2b × h^2^)(1)
where R is the bending strength (MPa), F is the failure load (N), L is the length of span (mm), b is the width (mm), and h is the thickness (mm); L = 250 mm, b = 50 mm, h = 12 mm in this research.
E = l_1_/lbt × [(F_2_ − F_1_)/(a_2_ − a_1_)](2)
where E is modulus of elasticity (MPa), l_1_ is the length of span (mm), l is the length of test piece (mm), b is the width (mm), t is the thickness (mm), F_2_ − F_1_ is the load increase (N), a_2_ − a_1_ is the increase in specimen deformation (mm); l_1_ = 230 mm, l = 250 mm, b = 50 mm, t = 12 mm in this research.

### 2.3. Thickness Swelling (TS) Test

The specimen was cut into a 50 mm × 50 mm × 12 mm piece, the thickness of the specimen was measured and recorded as T_1_ before immersion in water. The distance between the specimen and the water surface should not be less than 10 cm when immersed in the water. After 24 h, the specimens were taken out and wiped free from water to record the thickness as T_2_. The formula for TS is shown as below.
T = (T_2_ − T_1_)/T_1_ ×100%(3)
where T is thickness swelling (%), T_1_ is the thickness before immersing in water (mm), T_2_ is the thickness after immersing in water (mm).

### 2.4. Characterization

The specimens were put into an oven for drying treatment, the area near the interface between the wood and magnesium cement was cut into a 2 mm × 2 mm × 1 mm size. The test-pieces were pasted on the bench with conductive adhesive and sprayed with metal for field emission scanning electron microscopy (FESE, JSM-7800 F Prime, Japan Electronics, Tokyo, Japan) testing. The equipment vacuum pressure was 5 × 10^−4^ Pa. Resolution was 0.7 nm (15 KV). Acceleration voltage was 0.001–30 kV. Amplification factor was 10–10^6^ times.

The test pieces were crushed to 120 meshes and pressed with potassium bromide coating for Fourier transform infrared spectrometry (FTIR, Nicolet 6700, Thermo, Waltham, MA, USA) testing. The scanning range was 600–4000 cm^−1^, the scanning times were 32, the resolution 4 cm^−1^.

The test pieces were crushed to 120 meshes for the thermogravimetric analysis (TG, TAQ-50, TA Instruments, New Castle, DE, USA) test. Heating temperature was 20–600 °C, balance sensitivity was less than 1.5 µg; vacuum degree was less than 400 Pa, 99.99% nitrogen was used as protective gas and purge gas, air flow setting was 20 mL per min.

According to ISO 5660-1-2002, the specimens were prepared of 10 cm × 10 cm × 1.2 cm and wrapped with tin paper at the bottom for the cone calorimetry (CONE, NLFRM-05, Fire Testing Technology limited, East Grinstead, UK) test. The test temperature was from 20–1800 °C, thermal power was 50 kW/m^2^.

The test piece was crushed to 60 mesh and dried for the X-ray diffraction analyzer (XRD, Bruker D8, Karlsruhe, Germany) test. Scanning angle was 0–80°, the maximum voltage of the light tub was 40 kV, the maximum current of the light tube was 40 mA.

X-ray photoelectron spectroscopy (XPS, Escalab 250 Xi, Thermo Fisher Scientific, Waltham, MA, USA) was used to analyze the chemical composition changes of the elements. The samples were prepared of 10 mm × 10 mm × 0.5 mm size, the hemispherical energy analyzer was A1 target, the X-ray spot was 100 µm, the step was 0.1 eV, the incidence angle was 45°, the vacuum of the analysis chamber was 1.0 × 10^−7^ Pa, the reference for correction was C1s = 284.8 eV.

## 3. Results

### 3.1. Physical Combination of MCPB

The SEM images of MCPB are shown in Figure 2. Figure 2a–c have the same observation area with different scale. Meanwhile, Figure 2C1–C3 were selected from a different area of Figure 2c. The magnesia cement has a needle-like or rod-like structure which is inserted in the pits and aperture of the poplar. The micro-bonding morphology indicated that magnesia cement had entered the wood pits like “nails”. It revealed that the poplar and magnesia cement were combined in the form of “glue nails” and were interwoven with each other. The “glue nail” was a physical combination which formed a stable structure. The conclusion could be evidenced by Energy Dispersive Spectrometer (EDS) images as Figure 3 shows. The distributions of magnesium element (Mg) were colored green, the oxygen element (O) blue, and the carbon element (C) cyan. The hole was a pit of the poplar, the area around it was the cell of the poplar (cyan for the carbon element of wood), the rod-like material in the hole was magnesium cement (Figure 3 SEM), which is consistent with the color distributions of Mg and O, it proved the material in the pit was precisely magnesium cement.

### 3.2. Chemical Combination of MCPB

Figure 4 shows the FTIR result of MCPB. The peak of the hydrogen bond (O–H) and the amino (N–H) functional groups appeared at 3691 cm^−1^, the absorption peak of hydroxyl functional groups (–OH) appeared at 3610 cm^−1^, the peak of amide (–NH) appeared at 3380 cm^−1^, the peak of double the bond of C=C appeared at 1589 cm^−1^, which mostly exists in wood lignin. At 1186 cm^−1^, C–O was dominant indicating the C–O–C functional group. It can be seen from the figure that the peak of the MC + SM group is slightly stronger than the others at 1186 cm^−1^, which indicates the C–O functional groups were increased in the SM group. Compared with other groups, the peak of MC + PEG group was shorter at 1081 cm^−1^, which indicated that the C–OH hydroxyl group was reduced, it may be caused by the reaction between –OH in PEG and the composition of the wood.

Figure 5 shows the XRD results of MCPB with various chemical additions. There were 5-phase crystallization peaks at 2θ = 11.7°, 21.5°, and 37°. The 5-phase crystallization was a 5.1.8 crystal structure, which is the characteristic structure of MCPB with the molecular formula of 5Mg(OH)_2_·MgCl_2_·8H_2_O, the chemical reaction is shown as Equation (4). The MgO crystallization peak was observed at 2θ = 43°. The peak of the NH_4_Cl crystal appeared at 2θ = 32° and 34°. The MC + SPI group had sharp crystallization peaks, which indicated higher crystallization content in this group. The amount of crystal content was related to the mechanical properties of the MCPB.
(4)$${5\mathrm{MgO}+\mathrm{MgCl}}_{2}{+13H}_{2}{O=5\mathrm{Mg}(\mathrm{OH})}_{2}\cdot {\mathrm{MgCl}}_{2}\cdot {8H}_{2}O$$

### 3.3. Mechanical Properties and Water Resistance of MCPB

#### 3.3.1. Mechanical Test-MOR and MOE

The results of the modulus of elasticity (MOE) and the modulus of rupture (MOR) tests are listed in Table 2. The maximum value of MOE for the KH560 group was 3224 MPa, which corresponded to the standard of P4 heavy-duty particleboard (3100 MPa) of national standard GB/T 17657-2013 and GB/T 4897-2015 (GBT). As Figure 6a shows, the order of influence on the MOE for various additions was 1% > 3% > 7% > 5% > 15%, and it was 7% > 1% > 3% > 5% > 15% on the MOR. With the increase of maleic anhydride (MAH) addition from 1% to 5%, the MOE and MOR decreased as Figure 6b shows. The MOE was 2588 MPa at 7% addition, which corresponded to GBT P3 heavy particleboard (2200 MPa). The maximum MOR was 7.33 MPa, which does not reach the GBT (10.5 MPa).

Figure 6c shows the results of the mechanical properties for the polyethylene glycol (PEG-400) group. From 1% to 7% addition, the strength of MCPB increased. The MOE was 3200 MPa and the MOR was 10.52 MPa at 7%. Both reached the GBT, which illustrated that the PEG improved the mechanical properties successfully. PEG is a high molecular polymer with chemical formula HO(CH_2_CH_2_O)_n_H. It contains a large amount of single-bonds and has excellent properties of water solubility. The interfacial interaction of MCPB was enhanced with the addition of PEG-400.

After added polyacrylic acid (PAA) in MCPB, the MOE and MOR showed a decreasing line from 1% to 7%. The maximum MOE was 3023 MPa at 1%, which is up to the GBT. The polyacrylic acid can improve the distribution and structure to enhance the mechanical properties of the materials. From the FTIR results, the –COOH functional group band appeared in the PAA group at 3380 cm^−1^, which proved that after PAA mixing, the oxygen-containing functional groups were increased and an esterification reaction appeared.

Figure 6d shows the results of MOE and MOR after SM (soybean meal) modification. With the increase of SM addition, the line shows an upward trend, which means the mechanical properties increased. The MOE was 3296 MPa and the MOR was 9.23 MPa at 3%, which reached the GBT of P4 particleboard. Because there were more than 20 kinds of amino acids distributed on the peptide chain of SM, which have strong activity and can be chemically adsorbed by hydrophilic groups. The –NH group band appeared at 3380 cm^−1^ from the FTIR results, the crystallization of NH_4_Cl was shown from the XRD results, which proved the amino group in SM reacted with the hydroxyl in the wood.

Figure 6f shows the results of the mechanical properties with the SPI group. On addition of 3%, the MOE was 5192 MPa, which exceeded 67.4% of the GBT (3100 MPa of P4 heavy-duty particleboard). The MOR was 17.72 MPa, which exceeded 18% of the national standard for P3 type (15 MPa), as well as exceeding 48% of common particleboard P1 (10.5 MPa). SPI is the powder from the bean with a high viscosity, plasticity, and elasticity. The protein in SPI could form a structure-like “network” to enhance the mechanical properties of MCPB. The results of FTIR showed the –NH group band at 3380 cm^−1^. The results of XRD showed the crystallization peak of NH_4_Cl. The crystals of NH_4_Cl were formed by the combination of chloride ions from MgCl_2_ and ammonium ions from soybean protein. It indicated that the SPI improved the mechanical properties successfully.

#### 3.3.2. Water Resistance—Thickness Swelling (TS) after 24 h

Table 3 shows the water resistance analysis of MCPB. The TS of MAH and the SPI group were less than 0, which showed the thickness was shrinking instead of expanding after immersion in water. This phenomenon caused by the secondary combination of unreacted components in magnesium cement, led to the shrinkage of cell space. The values of SM, KH560, and PEG group were less than 3%, which reached the requirement of GBT (≤16 MPa). TS is a significant criterion for dimensional stability, the results of TS demonstrate that the MCPB had a good performance on thickness swelling, especially for the SM and KH560 group (as Figure 7 shows).

#### 3.3.3. Effect on the Mechanical Properties of NaOH Treatment

The mechanical properties of MCPB made of shavings treated with NaOH were higher than the untreated one (as Figure 8 shows). After the NaOH treatment, the extracts in the wood including tannin, pigment, soluble mineral composition, and some monosaccharides were swollen and extracted. After immersion in NaOH, the pores of the wood became clear because the extract in the cell wall was dissolved. As a result, more magnesium cement could penetrate into the pits of the wood through physical infiltration. The treatment of NaOH solution improved the interface between the wood fibers and the magnesium cement. On the other hand, NaOH solution can swell and dehydrate the water and small molecules in wood cellulose. The result from XPS (Figure 9) showed the energy spectrum changes of C1s in wood composition before and after NaOH treatment. The peak areas of C–C/C=C, C–O and C=O/O–C–O of C1s were increased after NaOH treatment (Table 4), it demonstrated the swelling and dehydration in the cellulose of the wood. 

### 3.4. Thermal Stability and Fire Resistance of MCPB

#### 3.4.1. Thermal Stability

Thermal gravimetric analysis (TG) refers to the percentage of mass reduction of samples under heating conditions. DTG (Differential thermal gravimetric analysis) is the differential curve of TG, which can be used to investigate the stability of the mass and the rate of mass loss in the process of heating.

As shown in Figure 10 and Figure 11, both the TG and DTG curves can be divided into four stages. In the first stage, the curve of TG descended from 20 °C to 180 °C, which meant the rate of weight loss was higher. In this stage, the bound water and crystalline water in the cell wall evaporated gradually with the increase of temperature. In addition, some volatile compounds with low molecular weight escaped. The crystalline phase of 5.1.8 began to lose crystalline water gradually at 150 °C. At the same time, the curve of the DTG changed obviously. There were two peaks at 80 °C and 160 °C, which indicated the rapid reduction of weight. The addition of SM improved the thermal stability effectively because the rate of weight loss in the SM group was slower than that of the empty group (the black curve).

The second stage was from 180 °C to 290 °C. Because of undecomposed material, the rate of mass loss was unchanged. This meant the thermal reaction of the microstructure had not started yet, as well as the fact of no change of the DTG curve. In the third stage, the temperature increased from 300 °C to 400 °C. The mass loss of the TG decreased from 75% to 55%. It indicated that the internal chemical composition of the MCPB was beginning to decompose. The decomposition of Mg(OH)_2_ and 5.1.8 phase was speeded up at 400 °C, as the DTG curves showed two peaks at 300 °C and 400 °C. The fourth stage was from 400 °C to 600 °C. The TG curve decreased slowly from 55% to 50% until the mass was stable. This stage was the end of the thermal reaction. The decomposition of the sample was complete. From the TG result, the final thermal decomposition mass of MCPB was 50%. The DTG line remained stable after 400 °C indicating that the thermogravimetric reaction was finished.

#### 3.4.2. Fire Resistance—Results of CONE

The heat release rate (HRR) is an important parameter to determine the scale of fire development. The larger the peak value, the faster is the pyrolysis of the material. As Figure 12 shows, the HRR of the PEG group increased abruptly at 800 s. The HRR of the KH560 group was lower than the empty group, which indicated a slower heat release and shorter time for the heat transfer of the KH560 group. After immersion in NaOH (the shavings in the KH560 group had been treated by NaOH before), the extracts of the wood dissolved, such as fat, tannin, sugar, and other substances. The interstitial space between the cells and the interstitial pathway were much clearer, as a result, more magnesium cementitious material entered the interstitial space of the wood cells. The 5-phase crystal from the magnesium material was stable and hard to burn, which could stop the decomposition of wood and slow down the combustion.

The total heat release (THR) is the value of heat release per weight unit of combustion, it is one of the important indicators to evaluate the flame retardancy of materials. From Table 5, the THR of the empty group reached 39.39 MJ/kg. The THR of the PEG group and KH560 group were 34.39 MJ/kg and 29.71 MJ/kg respectively, compared with the empty group, the value decreased by 12.7% and 24.6%, the THR reached the Chinese national standard GBT 8624-2012 (THP ≤30 MJ/kg). The result showed that PEG and KH560 could improve the flame retardancy of the material significantly.

The total smoke produced (TSP) of the PEG Group was 0.366 m^2^, the values of the other two groups were 0.192 m^2^ and 0.167 m^2^ about 50% less than the PEG group. Overall, MCPB produced lower smoke. The release rate of CO_2_ (CO_2_PR) and CO (COPR) in KH560 group were lower than the other two groups. The carbon emissions of the three kinds of magnesia cementitious composite particleboard were much lower than that of common wood particleboard (0.2898 g/s). 

The effective heat of combustion (EHC) is the ratio of material heat release and mass loss at a certain time. The higher the EHC value, the more is the heat released from the combustion of the unit. The EHC of the three groups can be observed in Figure 13. In the PEG group, the first EHC peak appeared at 65 s, with a value of 55.4 MJ/kg, and the other two peaks appeared at 310 s and 535 s, with EHC values 44.15 MJ/kg and 59.19 MJ/kg. The EHC value of the MC group was lower than the PEG group. In the SM group, the first EHC peak appeared at 345 s with the EHC value 50.79 MJ/kg, and the peak of EHC appeared at 600 s with the value 72.55 MJ/kg. The group of KH560 and MC had a better performance than the PEG Group. The CONE test was according to ISO 5660-1-2002 [21].

## 4. Discussion

In this research, MCPB was prepared and the mechanical strength, water resistance, thermal stability and flame retardancy were tested. The results showed that MCPB has excellent properties. Future research should focus on particleboard building materials constructed with magnesium cement.

## 5. Conclusions

Magnesium cement materials enter the pits in the wood as physical “glue nails”. After NaOH treatment, extracts of the wood were removed and the cell gap became clearer so much more magnesia cementitious material entered into the pits of the wood.

The results of FTIR showed that hydroxyl –OH and –NH functional groups appeared at 3600 cm^−1^ waveband. The XRD results showed the 5.1.8 phase crystallization and NH_4_Cl crystallization.

SPI addition (3%) improved the mechanical property of MCPB effectively. The MOE up to 5192 MPa, which exceeded the GBT 4897-2015 by 67.4% of P4 type (3100 MPa). The MOR was 17.72 MPa, which exceeded the GBT for P3 type (15 MPa) by 18%, exceeding the national standard of P1 bearing particleboard (10.5 MPa) by 48%. Thickness swelling in water after 24-hour was 0.29% (KH560 group), which reached the standard of the GBT (≤16%).

The MCPB had good thermal stability and reliability. The maximum thermal decomposition residual was 55.82% (SM group), and the minimum rate was 50.72% (PAA group).

The fire resistance test (CONE) showed that the KH560 group had good performance on flame retardancy. The minimum HRR was 18 kW/m^2^. The minimum TSP was 0.192 m^2^. The minimum value of THR was 29.71 MJ/kg, both reached the Chinese national standard GBT 8624-2012 (THP ≤30 MJ/kg, HRR ≤200 kW/m^2^).

## Figures and Tables

**Figure 1 materials-12-03161-f001:**
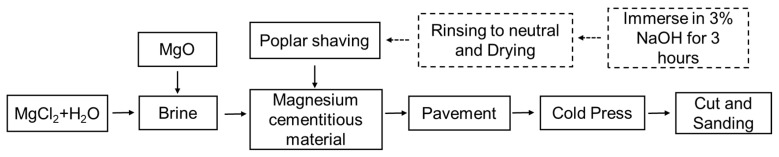
Production flow chart of MCPB.

**Figure 2 materials-12-03161-f002:**
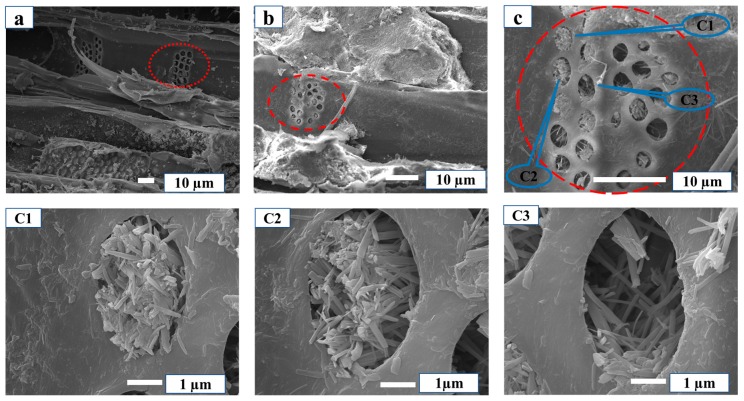
MC crossed like a nail in wood pit. (**a**) Pits of wood; (**b**) Pits of wood; (**c**) MC in the pits of wood; (**C1**) MC in the pits of wood; (**C2**) MC in the pits of wood; (**C3**) MC in the pits of wood.

**Figure 3 materials-12-03161-f003:**
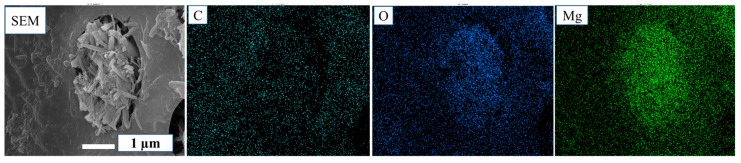
SEM and energy dispersive spectrometry (EDS) results.

**Figure 4 materials-12-03161-f004:**
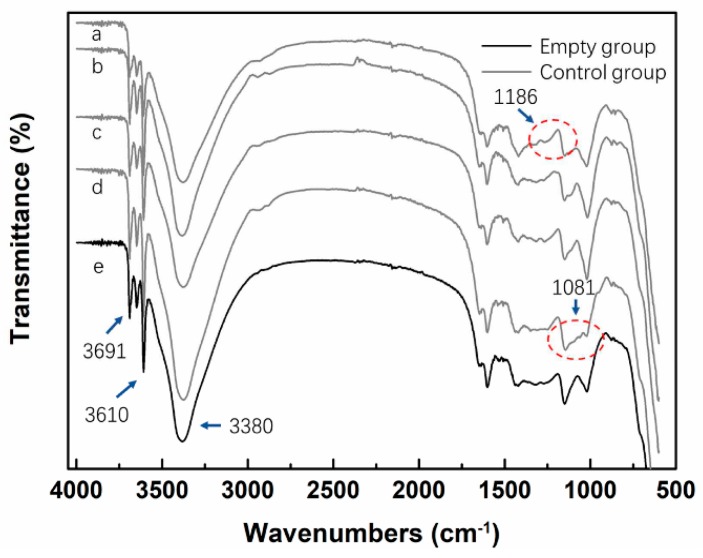
FTIR analysis for MCPB: (a): MC + SM group; (b): MC + KH560 group; (c): MC + PAA group; (d): MC + PEG group; (e): Empty group (MC without chemical addition).

**Figure 5 materials-12-03161-f005:**
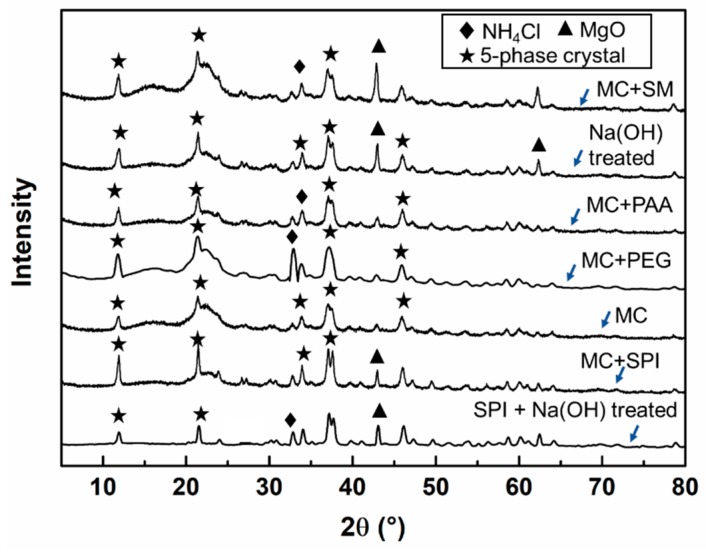
XRD analysis of MCPB.

**Figure 6 materials-12-03161-f006:**
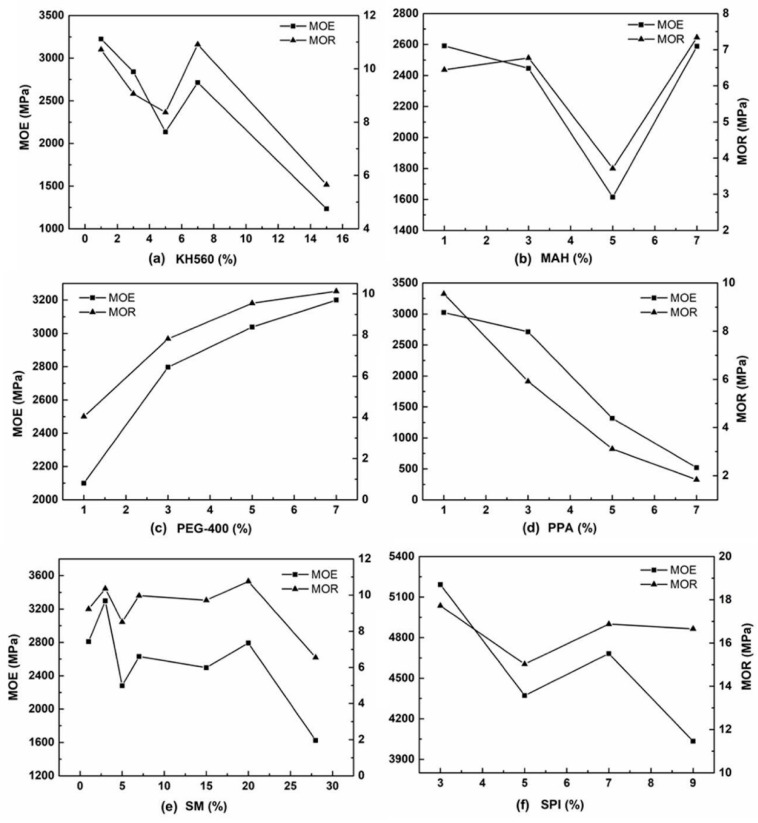
MOE and MOR for various chemical additives and different rates: (**a**) for KH560 addition; (**b**) for MAH addition; (**c**) for PEG addition; (**d**) for PAA addition; (**e**) for SM addition; (**f**) for SPI addition.

**Figure 7 materials-12-03161-f007:**
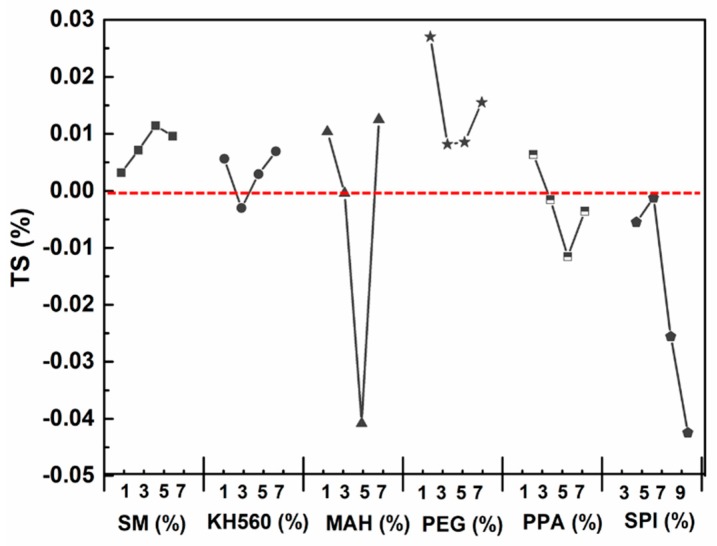
Value of TS after immersion in water for 24 h.

**Figure 8 materials-12-03161-f008:**
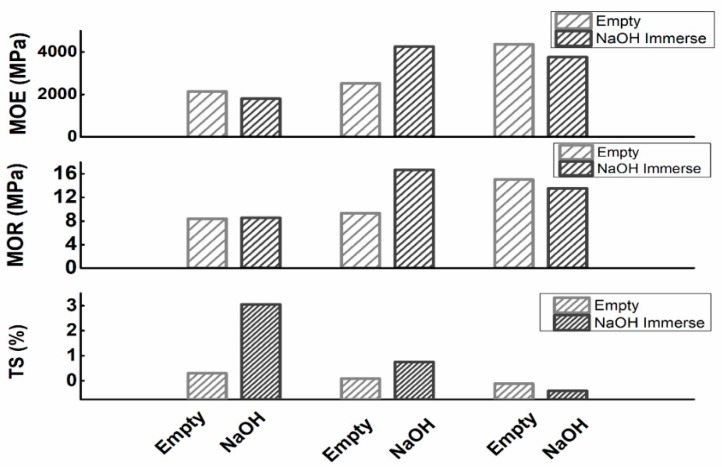
Comparison of MOE, MOR, and TS before and after NaOH immersion.

**Figure 9 materials-12-03161-f009:**
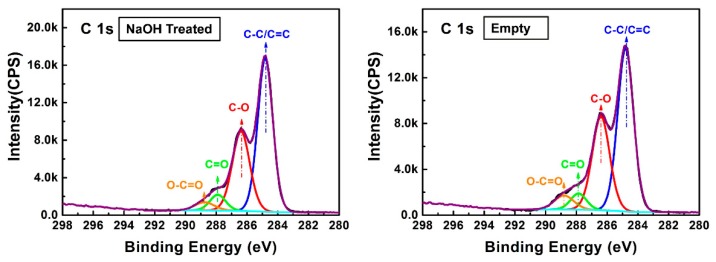
Comparison of C1s peak before and after NaOH treatment.

**Figure 10 materials-12-03161-f010:**
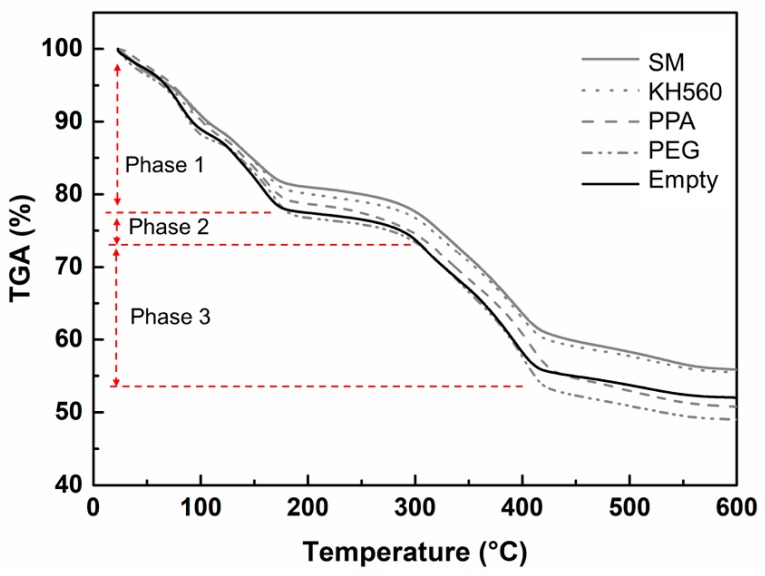
TG for MCPB with various chemical additions.

**Figure 11 materials-12-03161-f011:**
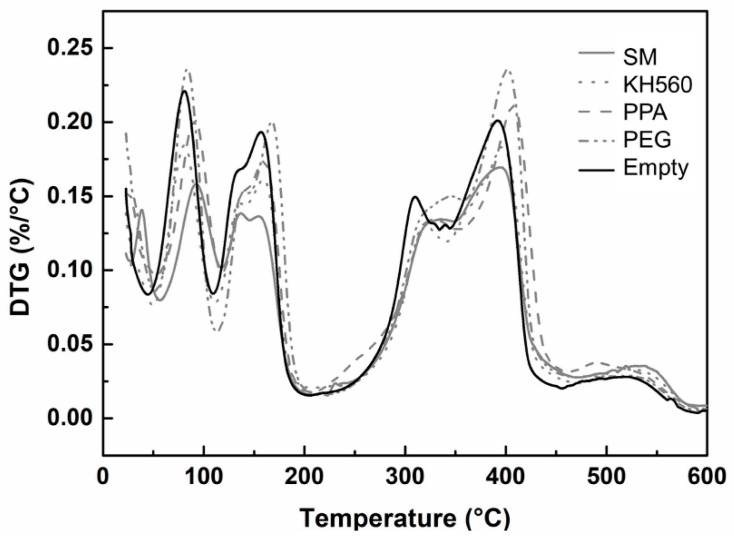
DTG for MCPB with various chemical additions.

**Figure 12 materials-12-03161-f012:**
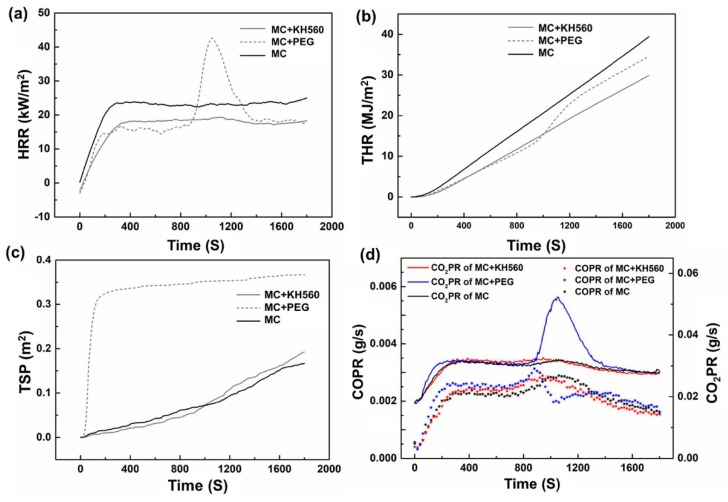
Results of CONE test. (**a**) Results of HRR; (**b**) Results of THR; (**c**) Results of TSP; (**d**) Results of COPR and CO_2_PR.

**Figure 13 materials-12-03161-f013:**
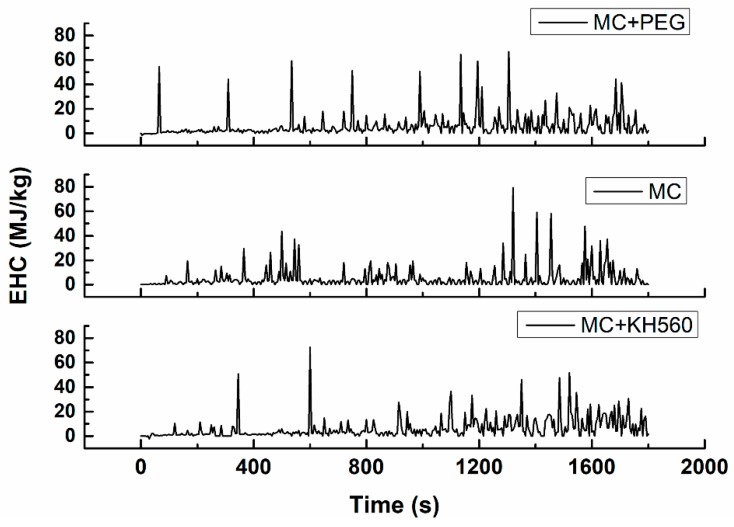
EHC of different chemical additions.

**Table 1 materials-12-03161-t001:** Raw material and manufacturer.

Material	Grade	Manufacturer
MgO	Industrial	Shi Jiazhuang Tian Yu Magnesium Industry Co., Ltd., Shijiazhuang, China
MgCl_2_	Industrial	Wuxi Yatai United Chemical Co., Ltd., Wuxi, China
H_2_O	Chemical	Laboratory of Beijing Forestry University
Soybean meal (SM)	Food grade	Shandong Yu Wang Ecological Food Co., Ltd., Dezhou, China
Silane coupling agent (KH560)	Chemical	Shanghai Macklin Biochemical Technology Co., Ltd., Shanghai, China
Polyacrylic Acid (PAA)	Chemical	Shanghai Macklin Biochemical Technology Co., Ltd., Shanghai, China
Polyethylene glycol (PEG-400)	Chemical	Shanghai Macklin Biochemical Technology Co., Ltd., Shanghai, China
Soybean protein isolate (SPI)	Food grade	Shan Dong Yu Wang Ecological Food Co., Ltd., Dezhou, China
Maleic anhydride (MAH)	Chemical	Shanghai Macklin Biochemical Technology Co., Ltd., Shanghai, China
NaOH	Chemical	Tian Jin Da Mao Chemical Reagent Factory

**Table 2 materials-12-03161-t002:** The modulus of elasticity (MOE) and the modulus of rupture (MOR) for various chemical additions.

Mechanical Test	Chemical Addition	Ratio (%)				
		1	3	5	7	9
MOE (MPa)	SM	2809.08	3296.69	2280	2630.69	—
	KH560	3224.22	2839.84	2134.43	2713.36	—
	MAH	2590.7	2445.78	1614.5	2588.31	—
	PEG	2099.83	2796.89	3037.93	3200	—
	PAA	3023.78	2713.15	1316.42	520.6	—
	SPI	—	5192.83	4371.5	4682.33	4033.5
MOR (MPa)	SM	9.23	10.36	8.51	9.97	—
	KH560	10.72	9.06	8.36	10.91	—
	MAH	6.44	6.77	3.71	7.34	—
	PEG	4.04	7.82	9.54	10.13	—
	PAA	9.55	5.92	3.11	1.84	—
	SPI	—	17.73	15.03	16.88	16.66

**Table 3 materials-12-03161-t003:** TS (%) for various chemical additions.

Proportion (%)	TS (%) with Various Chemical Additions					
	SM	KH560	MAH	PEG	PAA	SPI
1	0.32	0.56	1.04	2.7	0.64	−0.55
3	0.71	−0.3	−0.04	0.82	−0.15	−0.12
5	1.14	0.29	−4.08	0.86	−1.15	−2.56
7	0.96	0.69	1.25	1.55	−0.36	−4.25

Note: Red font means less than zero.

**Table 4 materials-12-03161-t004:** Peak fitting of C1s before and after NaOH treatment.

Samples	Parameter	Peak Fitting of C1s			
		C–C/C=C	C–O	C=O/O–C–O	O–C=O
NaOH treated	Peak position	284.8	286.4	287.9	288.8
	Peak area	20443.6	12273.5	2201.1	1421.6
Empty	Peak position	284.8	286.4	287.9	288.8
	Peak area	18759.1	11867.4	1838.2	1981.8

**Table 5 materials-12-03161-t005:** Results of CONE test.

Test Group	HRR (kW/m^2^)	TSP (m^2^)	THR (MJ/kg)	CO Release (g/s)	CO_2_ Release (g/s)
26(MC + KH560)	18	0.192	29.71	0.002	0.031
33(MC + PEG)	43	0.366	34.39	0.003	0.051
35(MC)	24	0.167	39.39	0.002	0.032

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
