# Peer review of "Mechanical Properties and Fire Resistance of Magnesium-Cemented Poplar Particleboard"

_materials, 2019, doi:10.3390/ma12193161_

Round 1
Reviewer 1 Report
The paper is addressed to rather limited number of readers, since the kind of the construction products and materials that are dealt with is not very widely applied as for now. Nevertheless, within this range the paper is well prepared and brings interesting new information, which can be useful in making this solution more popular. I recommend to publish the paper, but first the language used need to be corrected and carefully checked. There are a lot of errors of this type (missing words or grammar structures making the text hard to understand), which make the reading somewhat annoying. Some examples: "adhesives were consumed about 20 million tons", "environmental material" (what is it?), "Property ... were tested and compared by adding various chemical addition" (it is not possible to test property by just adding one material to another, you need to measure something). "40.31" is not the "molecular formula", which is MgO, but molecular mass. There is also definitely unwanted to insert a separate table containing detailed information about the producers of such equipment like moulds, steel plates or mortar mixer. Instead of this, it would be useful to give some brief information on the action of magnesium cement, since this material is not widely used today and so not every reader is familiar with this type of the binder.
Author Response
Reviewer 1:
The paper is addressed to rather limited number of readers, since the kind of the construction products and materials that are dealt with is not very widely applied as for now. Nevertheless, within this range the paper is well prepared and brings interesting new information, which can be useful in making this solution more popular. I recommend to publish the paper, but first the language used need to be corrected and carefully checked. There are a lot of errors of this type (missing words or grammar structures making the text hard to understand), which make the reading somewhat annoying. Some examples: "adhesives were consumed about 20 million tons", "environmental material" (what is it?), "Property ... were tested and compared by adding various chemical addition" (it is not possible to test property by just adding one material to another, you need to measure something). "
Thanks for your suggestion. I had corrected and checked the language in this article carefully.
"40.31" is not the "molecular formula", which is MgO, but molecular mass.
I have changed "molecular formula" to “molecular mass”.
There is also definitely unwanted to insert a separate table containing detailed information about the producers of such equipment like moulds, steel plates or mortar mixer. Instead of this, it would be useful to give some brief information on the action of magnesium cement, since this material is not widely used today and so not every reader is familiar with this type of the binder.
Thanks for your comments. I have deleted the table and added information about magnesium cement at line 22-25.
Reviewer 2 Report
What is the main aim of this work? Products composition or Mechanical Property or Fire Resistance?
If mechanical properties in witch temperature intervals? You declared very good mechanical properties (25 oC), but in TG curves we see ~ 50 wt. % mass loss (at 600 oC) and a good fire resistance test results, but what about mechanical properties in this temperature (600 oC)?
If products composition: what differences and similar between You pervious work: Zheng et. Al. (2019) Mg-cemented straw composites, BioResources 14 (3), 7285-7298, and others: Y. Zuo et al. Construction and Building Materials. 171, 2018 404-413; etc.
The quality of XRD curve is poor, used etalons; d-spacing; pdf number (add it in text). All peaks must be identified. Quantitative composition of products: The description of XRD analysis should be combined with thermal analysis and FT-IR results. Please to confirm this fact (FT-IR, TGA results) by literary data.
What was the principle to use to design an experiment? How do they complement / confirm each other? And which determined parameters is the imported for this type materials?
Author Response
Reviewer 2
What is the main aim of this work? Products composition or Mechanical Property or Fire Resistance?
The mechanical property, water resistance and flame retardancy were researched and evaluated synthetically in this study. Not all of them were very strong, but the materials were prepared under different weight vector and condition of use. If the mechanical property is the main considerations, the SPI group is suitable. If you consider the water resistance as main factor, the KH560 group will be more suitable.
If mechanical properties in which temperature intervals? You declared very good mechanical properties (25 oC), but in TG curves we see ~ 50 wt. % mass loss (at 600℃) and a good fire resistance test results, but what about mechanical properties in this temperature (600℃)?
The mechanical property is for the normal temperature, not for the temperature of 600℃. It cannot be measured in 600℃. Weight loss (in TG curve) is the standard for thermal stability, which is related to the use range of materials in the condition of fire, it’s different from mechanical property. This research studied the mechanical property and thermal stability at the same time to analysis and prepare materials suitable for different conditions of use.If products composition: what differences and similar between You pervious work: Zheng et. Al. (2019) Mg-cemented straw composites, BioResources 14 (3), 7285-7298, and others: Y. Zuo et al. Construction and Building Materials. 171, 2018 404-413; etc.
They are different products with different raw material, this article is for particleboard with popular and magnesium cement. But it’s for rice straw and fly ash composited with magnesium cement in the previous work and others. They are different method of solution. this article use chemical addition such as KH560, SPI, PAA, PEG, NaOH to improve property (mechanical, water resistance and fire resistance) for MCPB. But it’s no (or not the same) chemical addition for my previous work and others.The quality of XRD curve is poor, used etalons; d-spacing; pdf number (add it in text). All peaks must be identified. Quantitative composition of products:
I have updated the figure of XRD.
The description of XRD analysis should be combined with thermal analysis and FT-IR results. Please to confirm this fact (FT-IR, TGA results) by literary data.
The function group of -OH and -NH were found at 3380 cm-1 in FTIR. the crystal of NH4Cl and 5 phase crystal 5Mg(OH)2.MgCl2.8H2O were found in XRD results, which contains -NH and -OH. Both of the two analysis were consistent with each other.
What was the principle to use to design an experiment? How do they complement / confirm each other? And which determined parameters is the imported for this type materials?
The single factor experiment was designed in this research. The addition of chemical addition was the important parameters which was determined by the pre-experiment. We had arranged some pre-experiment to fix the range of addition from 1% to 9%, the results showed the MCPB got the very low MOE and MOR out of this range (1% to 9%).